# An Update on Recent Advances of Photodynamic Therapy for Primary Cutaneous Lymphomas

**DOI:** 10.3390/pharmaceutics15051328

**Published:** 2023-04-24

**Authors:** Wei-Ting Liu, Han-Tang Wang, Yi-Hsuan Yeh, Tak-Wah Wong

**Affiliations:** 1Department of Dermatology, Cancer Center, National Cheng Kung University Hospital, College of Medicine, National Cheng Kung University, Tainan 704, Taiwan; ly5508@hotmail.com (W.-T.L.); gaspard.ong@gmail.com (H.-T.W.); 2School of Medicine, National Cheng Kung University, Tainan 701, Taiwan; cynthia.yeh360@gmail.com; 3Department of Biochemistry and Molecular Biology, College of Medicine, National Cheng Kung University, Tainan 701, Taiwan; 4Center of Applied Nanomedicine, National Cheng Kung University, Tainan 701, Taiwan

**Keywords:** photodynamic therapy, primary cutaneous lymphoma, CTCL, mycosis fungoides, CBCL

## Abstract

Primary cutaneous lymphomas are rare non-Hodgkin lymphomas consisting of heterogeneous disease entities. Photodynamic therapy (PDT) utilizing photosensitizers irradiated with a specific wavelength of light in the presence of oxygen exerts promising anti-tumor effects on non-melanoma skin cancer, yet its application in primary cutaneous lymphomas remains less recognized. Despite many in vitro data showing PDT could effectively kill lymphoma cells, clinical evidence of PDT against primary cutaneous lymphomas is limited. Recently, a phase 3 “FLASH” randomized clinical trial demonstrated the efficacy of topical hypericin PDT for early-stage cutaneous T-cell lymphoma. An update on recent advances of photodynamic therapy in primary cutaneous lymphomas is provided.

## 1. Introduction

Lymphomas are cancerous disorders that start from lymphocytes, consisting of varying groups that are generally divided into Hodgkin and non-Hodgkin lymphomas. They can arise from the lymph nodes, spleen, bone marrow, or extranodal organs including the skin [1]. Primary cutaneous lymphomas (PCLs) are a heterogenous group of extranodal non-Hodgkin lymphomas of T-cell, NK-cell, or B-cell origin, which primarily affect the skin. PCLs are uncommon, accounting for only about 4% of all non-Hodgkin lymphomas [2]. The classification and treatment guidelines of PCLs is mainly from the World Health Organization (WHO), updated in 2018, which is mainly based on the cell types (T-cells or B-cells), histopathology, and certain proteins expressed on the tumor cell. Aside from the up-to-date National Comprehensive Cancer Network (NCCN) clinical practice guidelines including diagnosis and treatment of primary cutaneous lymphomas [1], guidelines across different societies, aiming at diagnosis, staging, or management of PCLs are evolving and are summarized in Figure 1.

In the last decade, the 2005 World Health Organization-European Organization for Research and Treatment of Cancer (WHO-EORTC) consensus classification has served as a golden standard for the diagnosis and classification of PCLs [3]. According to the updated version published in 2018, the number of disease entities has increased, while the frequency of each subtype other than mycosis fungoides has been quite low [4]. The diagnosis of PCLs is based on clinical, histological, immunophenotypical, and molecular information. Systemic lymphoma or lymph node metastasis to the skin is excluded from PCLs even though they have similar histological features, because of the substantial differences in their management and prognosis [5,6]. The diagnosis of PCLs involves initial identification by dermatologists and histopathological evaluation of skin biopsy. Symptoms of PCLs are dependent on its classification of specific disease subtype, which may include lesions accompanied by fatigue, weight loss, or fever [5,6].

Photodynamic therapy (PDT) is a non-invasive therapy with the advantages of high tumor selectivity, good to excellent cosmetic outcomes, a large treatment field, and repeatability of the treatment. PDT was discovered by Professor Herman von Tappeiner and his student, Oscar Raab, in 1904 [7,8]. The term PDT was later coined by Von Tappeiner in his book, indicating that oxygen is required for the phototoxicity of cells. The modern history of PDT began with the first systemic clinical trials for skin cancers led by Thomas Dougherty at the Roswell Park Cancer Institute in the 1970s, which eventually led to FDA approval of this procedure [7].

The three key elements of PDT include a photosensitizer (PS), light, and oxygen [9]. After activation with an appropriate wavelength of light (usually visible light), PS is excited to an excited singlet ^1^PS (Figure 2). The excited ^1^PS forms a relatively long-lived triplet state ^3^PS by intersystem crossing and transfer of its electrons (Type I reaction) to neighboring substrates such as lipid, protein, RNA, DNA, or oxygen to produce reactive oxygen species (ROSs), including the superoxide anion radical (O2^•−^), hydrogen peroxide (H_2_O_2_), and the hydroxyl radical (OH•). It can also transfer its energy directly to oxygen to generate singlet oxygen (Type II reaction) in a cell. Note that the Type II reaction is believed to be the major reaction in killing a cancer cell or a pathogen [9]. The PS tends to accumulate in cancer cells after a local or systemic application of a PS due to the different physiological and vascular components of the cancerous tissue compared to healthy tissue [9].

For the PDT of superficial skin cancers such as actinic keratosis and superficial basal cell carcinoma, a light dose of 37 J/cm^2^ was used widely in early clinical trials [10]. Moreover, even a very low accumulated light dose (3.5–8 J/cm^2^) has shown its effectiveness in daylight PDT for AK [8]. Nevertheless, the light doses required for effective PDT on caners are usually above 100 J/cm^2^ with red light, while in treating benign lesions, such as acne, or to improve wound healing, they are usually around 10–50 J/cm^2^ [11]. The formula for the light dose calculation is stated as follows: light dose (J/cm^2^) = light intensity (mW/cm^2^) × irradiation time (seconds). The most-common topical PS used today in dermatology is aminolevulinic-acid (ALA)-based, such as a 20% ALA solution (Levulan Kerastick^®^, Sun Pharmaceutical Industries Ltd., Mumbai, India) and a 10% gel form of ALA (Ameluz, Biofrontera AG, Leverkusen, Germany) for non-melanoma skin cancers [10]. ALA is a prodrug that metabolizes into protoporphyrin IX (PpIX), a real PS, which is selectively accumulated in neoplastic cells. In addition to treating skin cancers, ALA is currently used for fluorescence-guided surgical resection and PDT of high-grade gliomas and other brain tumors [12]. Coherent light sources (laser) and incoherent light sources have been widely used to activate different PSs in PDT. Lasers have the benefit of utilizing a high light intensity that can reach the required effective light dose within seconds to minutes. However, the major drawback of lasers is their high cost, as well as their small treatment field. They are usually used to treat internal malignancy such as cervical cancer, lung cancer, oral cancer or, brain cancer, etc. [13]. Incoherent light sources including xenon lamps, metal halide lamps with filters, and light-emitting diodes (LEDs) have been used to treat skin cancers. Among these light sources, LEDs have been widely adopted in PDT because of their cost-effectiveness, very low heat generation, relatively narrow range of wavelengths, and a large illumination area. That being said, blue light LEDs with a peak Soret band at 410 nm, the major absorption peak of PpIX, and red LEDs peaking at 635 nm are the most-common light sources for PDT in dermatology. A longer wavelength penetrates deeper into the skin; therefore, red light is usually used in treating cutaneous lymphoma. Nevertheless, the depth of red light penetration in the skin is less than 6 mm. Thus, ALA-PDT is limited to treating thin or stage I cutaneous lymphoma. Silicon phthalocyanine 4 is also a porphyrin-based PS, with an absorption peak at 675 nm, which can treat a thicker and deeper tumor in the skin. A non-porphyrin PS, hypericin, is a naturally occurring PS mainly existing in the plant Hypericum perforatum (St. John’s Wort) with absorption peaks at 545 nm and 590 nm [14]. A synthetic hypericin (HyBryte^®^, 0.25% ointment, Soligenix, Inc., Princeton, NJ, USA) has gained much attention recently due to the promising results in treating skin lesions of cutaneous T-cell lymphoma in phase 3 clinical study. PDT destroys neoplastic cells via the generation of singlet oxygen and reactive oxygen species (ROSs) during light activation of a PS, not related to the thermal damages. However, indocyanine green (ICG), a fluorescence imaging dye in hepatic, cardiac, and ophthalmologic perfusion examinations, had been used as a PS with absorption peaks in the near-infrared (NIR) (600–950 nm). The NIR penetrates deeper into the skin and generates heat during irradiation. ICG-mediated PDT has been regarded as a photothermal therapy (PTT) [15]. Either a dye-based PS such as methylene blue and ICG or porphyrin-based PS such as ALA, the dye or fluorescence fades (photobleaching) after irradiation, which can be interpreted as the degradation of the PS in tissues or cells.

PDT was generally used to treat skin cancer initially. However, with its unique cell-killing mechanism by generating singlet oxygen and ROSs and high selectivity in neoplastic cells, it is now widely used in the treatment of different cancers [13]. No PDT-resistant cells have been reported so far. The exclusion mechanism of PDT leads to the exploration of studies using PDT in the microbiology field [15,16,17]. However, light penetration is the major challenge in treating infectious diseases such as systemic infection and treating an abscess deep in the skin or an organ. It should also be noted that PDT is limited to treating thin and superficial cutaneous lymphoma.

The survival rates for PCLs differ significantly depending on the type of lymphoma, staging, and their responses to treatment [6]. In addition, based on the staging of the disease, skin-directed therapies (SDTs) for PCLs include topical corticosteroids, imiquimod, resiquimod gel, mechlorethamine solution or ointment, carmustine, bexarotene gel, tazarotene cream or gel, or radiotherapy. On the other hand, systemic treatments include methotrexate, chemotherapy, target therapy, immunotherapy, and phototherapy with broadband or narrowband ultraviolet B (UVB), psoralen and UVA (PUVA), or total skin electron beam therapy (TSEBT) applied to widespread skin lesions [6]. However, the effectiveness of the traditional treatments is mostly limited to symptom reduction with spaces for improvements in terms of their safety and maintenance of the quality of life. With this, it is of interest to further evaluate the application of PDT as an effective treatment tool in this scenario. Therefore, this article aims to address and discuss the efficacies of PDT with respect to PCLs in detail.

## 2. Primary Cutaneous Lymphomas

### 2.1. Classification of Different Subtypes of Cutaneous Lymphoma

Most of the PCLs are cutaneous T-cell lymphomas (CTCLs) (75–80%), and around 20–25% are cutaneous B-cell lymphomas (CBCLs) with different subtypes in the Western world (Table 1). Such differences are even greater in the Asian population, where up to 85.7% are CTCLs, as demonstrated in a Japanese nationwide study [2,6,18]. Epidemiological studies have shown that rare subtypes of CTCLs are more frequently found in Asian countries, whereas other types are evenly spread in other regions across the world [2].

#### 2.1.1. Cutaneous T-Cell Lymphoma

Around half of all CTCLs are mycosis fungoides (MF) [20]. MF affects mostly men and is twice as frequent compared to women in their 50s or 60s. The early stage of MF can be confused with other dermatoses with itchy or asymptomatic patchy, scaly, and red lesions on the skin, which poses the need for several skin biopsies to confirm. The disease course is usually slow and can take years to develop into plaques and tumorous stages. Some MF patients may develop Sezary syndrome (SS) [21]. Rare variants of MF include folliculotropic MF, pagetoid reticulosis, and granulomatous slack skin [22]. SS typically affects all skin with redness and severe itching, so-called erythroderma [22]. The tumor cells are characterized by cerebriform nuclei found in the blood, skin, and lymph nodes. Compared to MF, SS is harder to treat and grows faster, and patients are at a higher risk of serious infection with weakened immunity [22].

Adult T-cell leukemia-lymphoma (ATLL) affects mainly other parts of the body, but can sometimes be limited to the skin. ATLL is linked to human T-cell leukemia virus Type 1 (HTLV-1) infection, which often occurs in HTLV-1-endemic areas, such as Japan, the Caribbean islands, Central and South America, Intertropical Africa, and the Middle East [23]. This type of lymphoma is usually aggressive, but may grow slowly in some cases or even resolve over time.

Primary cutaneous anaplastic large-cell lymphoma (C-ALCL) usually affects people in their 50s or 60s, but can also occur in children. It begins with one or a few tumors of various sizes with or without ulceration on the skin [24]. The prognosis of this lymphoma is very good.

Lymphomatoid papulosis (LyP) is a benign, waxing and waning, slow-growing disease, even without treatment [25]. However, it might progress to lymphoma. The histology of LyP shows features that look like primary cutaneous ALCL. LyP affects younger men more than women, with an average age of 45 years old. The disease usually begins as several large pimple-like lesions with a central ulcer. Although it is rare, some LyP patients may develop a more serious type of lymphoma [25].

Extranodal NK/T-cell lymphoma, nasal type, usually begins in the nose or sinuses, but sometimes begins in the skin [26]. It is related to Epstein–Barr virus (EBV) infection, which is more common in Asia, as well as Central and South America. The tumor grows fast and aggressively [26].

There are several rare subtypes of primary cutaneous peripheral T-cell lymphoma, which are described in the following paragraphs. 

Primary cutaneous gamma/delta T-cell lymphoma usually develops as thick plaques or tumors on the skin of the arms and legs, but sometimes on the nasal mucosa or in the intestines [27]. This lymphoma grows and spreads quickly.

Primary cutaneous CD8+ aggressive epidermotropic cytotoxic T-cell lymphoma (CD8+ AECTCL) develops as widespread patches, nodules, and tumors with a central ulcer. This lymphoma may sometimes look like mycosis fungoides clinically, but tends to grow and spread quickly [28].

Primary cutaneous acral CD8+ T-cell lymphoma is very rare and typically grows on the acral skin such as the ear, nose, hand, or foot. It grows slowly and can be cured after treatments [29].

Primary cutaneous CD4+ small/medium T-cell lymphoproliferative disorder often begins as a single plaque or tumor on the head and neck area or the upper trunk. This lymphoma grows slowly and can be cured with treatments [30].

#### 2.1.2. Cutaneous B-Cell Lymphoma

PLCs with B-cell origin are referred to as cutaneous B-cell lymphomas (CBCLs). Different types of CBCLs are primary cutaneous marginal zone B-cell lymphoma, primary cutaneous follicle center lymphoma, and primary cutaneous diffuse large B-cell lymphoma, leg type [31,32].

Primary cutaneous marginal zone B-cell lymphoma is a very slow-growing, curable lymphoma which is sometimes caused by Borrelia infection in Europe [33]. The disease can affect people of all ages with more cases in older adults. The tumor can appear as red to purplish, with single or a few papules, plaques or nodules on the arms or upper trunk [34].

Primary cutaneous follicle center lymphoma is the most common, slow-growing B-cell lymphoma of the skin in middle-aged adults [35]. They appear as groups of red papules, nodules, or plaques on the scalp, forehead, or upper trunk, and sometimes on the legs. These tumors are sensitive to radiation and the outcome of the disease is excellent [36].

Primary cutaneous diffuse large B-cell lymphoma, leg type, is a fast-growing lymphoma that starts as large nodules on the lower legs in older people, affecting more women than men [37]. The tumor requires intensive treatment and may spread to lymph nodes and internal organs to cause serious disease. In general, a single lesion at the time of diagnosis has a better prognosis [37].

### 2.2. Management of Cutaneous T-Cell Lymphomas

Early aggressive treatments with combined chemotherapy and radiation therapy provide no overall survival benefit in patients with mycosis fungoides [38], and cures, if any, are rarely achieved [39]. The general consideration for the treatment of mycosis fungoides is aiming at achieving disease control while balancing quality of life at the same time, with treatment modalities tolerated for a longer duration and less cumulative toxicity. SDTs are suitable especially for patients with early-stage (stages IA-IIA) mycosis fungoides [40]. For patients with limited skin involvement, topical therapies including topical corticosteroids, topical mechlorethamine (nitrogen mustard), topical carmustine (BCNU), topical retinoids (bexarotene, tazarotene), and topical imiquimod are all optional local treatments [41]. The current guideline from National Comprehensive Cancer Network (NCCN) suggests no priority in choosing either one of the SDTs. Phototherapy with narrow-band UVB, PUVA, or TSEBT should be considered for patients with generalized skin involvement [1].

For primary cutaneous CD30+ T-cell lymphoproliferative disorders, which encompass primary cutaneous anaplastic large-cell lymphoma (ALCL) and lymphomatoid papulosis (LyP), involved-site radiation therapy or surgical excision are options for solitary or grouped lesions of primary cutaneous ALCL, while topical steroids or phototherapy are suitable for limited lesions of LyP. For all other rare subtypes of CTCLs, treatments depend on each subtype, and SDTs are suggested only for those with indolent behavior. Excision and/or local radiotherapy are recommended for CD4+ small/medium pleomorphic T-cell lymphoproliferative disorder and primary cutaneous CD8+ acral T-cell lymphoma by the British Association of Dermatologists and U.K. Cutaneous Lymphoma Group Guidelines [40]. For the treatment of CTCLs, photodynamic therapy is mentioned only in the British guidelines for mycosis fungoides, serving as an alternative for solitary plaques that are resistant to topical treatment.

### 2.3. Management of Cutaneous B-Cell Lymphomas

For primary cutaneous follicle center lymphoma (PCFCL) or primary cutaneous marginal zone lymphoma (PCMZL), SDTs including low-dose localized radiation therapy, topical or intralesional steroids, or even the watch-and-wait strategy are excellent treatment options for solitary/regional (T1-2) or even generalized skin-only (T3) disease [42]. Whereas for the aggressive variant primary cutaneous diffuse large B-cell lymphoma (PCDLBCL), the treatment rationale is similar to systemic diffuse large B-cell lymphoma, where localized radiation therapy is added to chemoimmunotherapy. However, none of the society guidelines recommend photodynamic therapy as a treatment option for CBCLs [43].

## 3. The Role of Photodynamic Therapy in Treating Primary Cutaneous Lymphomas

SDTs are treatments applied externally on the skin, with advantages including ease of use and minimal side effects compared to systemic therapies [40]. SDTs include various topical therapy, phototherapy (either UVB or PUVA), or radiation therapy and serve as monotherapy in early-stage PCLs or as adjunctive treatments in advanced-stage PCLs [1,44]. Randomized trials are lacking for both CTCLs and CBCLs. Most of the studies are case reports or limited clinical trials for mycosis fungoides, and studies on CBCLs are even fewer. The results are difficult to compare because of the different outcome measurements in these reports [41,45]. However, it is interesting to note that, recently, Caccavale et al. reviewed PDT for MF in 60 patients with 81 patch/plaque lesions and one tumorous lesion with a complete response rate that ranged from 20–100% [46]. Intriguingly, the lesions of the tumor stage MF patient achieved complete response at 60 months follow-up after three sessions of methyl-aminolevulinate PDT [47]. Moreover, in another study, Hooper et al. reported a 67% complete response rate in 44 stage IA MF with 55 lesions in a recent review [48]. That being said, we treated patients with promising results who suffered from early-stage MF and lymphomatoid papulosis with ALA-PDT (Figure 3, unpublished data). The first report of successful ALA-PDT treatment of three cases with early CBCL dates back to 2006 [49]. Recently, Toulemonde et al. reported a case series of four patients with marginal-zone-lymphoma (MZL)-type CBCL successfully treated with adjuvant PDT (Figure 4) [50].

### 3.1. History of PDT for Primary Cutaneous Lymphomas

Photodynamic therapy (PDT) is a type of treatment where visible light is used to penetrate the skin, directed towards the target cells that have photosensitizing substances accumulated [51]. The visible light allows for the selective destruction of the target tissues, which are usually malignant tumors or precancerous cells [51]. The specific mechanisms of PDT in treating CTCLs are still poorly understood, especially in the areas of the responses from inflammatory cells during the treatment process, as well as PDT’s contribution to neoplastic cell death in CTCL [46]. Despite this, past works have shown beneficial clinical effects while remaining tolerable for patients [48]. The successful application of PDT towards mycosis fungoides (MF), a subtype of CTCL, was first reported in 1994 by Wolf et al. (Figure 5) [46,52]. Since then, many other case reports also demonstrated successful outcomes [48]. For example, the work conducted by Edstrom et al. demonstrated significant clearance of local plaque lesions with the application of 5-ALA as the main photosensitizer [51]. In addition, Vallecorsa et al. made improvements by utilizing ALA derivatives in the treatment of MF with PDT [53]. Overall, as shown in most studies, PDT has shown its effectiveness in aiming at malignant cutaneous T-cells selectively in the treatment of CTCL.

### 3.2. In Vitro Studies of PDT for PCLs

The effects of PDT on lymphocytic cells with different photosensitizers have been intensively studied; however, studies on the optimal light dose and the exact mechanism of cell apoptosis are still limited. Subcellular localization of photosensitizers is crucial to photodynamic therapy efficacy and is dependent on the type and concentration of the photosensitizer, as well as the cell line used. Studies using different lymphoma cell lines demonstrated subcellular organelle uptakes with different photosensitizers (Figure 2). Trivedi et al. used confocal fluorescence microscopy to identify the uptake of silicon phthalocyanine (Pc 4) in mouse lymphoma (LY-R) cells [54]. They found that Pc 4 binds preferentially and strongly to the mitochondria and Golgi apparatus. Ke et al. identified Pc 4 uptake in both mitochondria and endoplasmic reticulum in human T lymphocytes (Jurkat cells) and epidermoid cells (A431 cells), with Jurkat cells much more sensitive to Pc 4-PDT than A431 cells [55]. On the other hand, 5-ALA-induced protoporphyrin IX (PpIX) was primarily localized in the endoplasmic reticulum and mitochondria, while much lower in the lysosome and in follicular lymphoma DHL cells, as demonstrated by two-photon excitation fluorescence [56]. Hexaminolevulinate (HAL), a hexylester of ALA, also targeted mitochondria and endoplasmic reticulum and induced apoptosis in two human lymphoma cell lines [57].

In vitro phototoxicity studies showed that PDT can induce cell apoptosis in different lymphoma cell lines. Caspase-dependent apoptosis was evidenced by HAL-PDT in Namalwa and Bjab lymphoma cell lines [57], while HAL-PDT in Jurkat T-cells demonstrated both the cytochrome c-mediated caspase-dependent pathway and the apoptosis-inducing-factor (AIF)-induced caspase-independent pathway [58]. A change in the anti-apoptotic protein Bcl-2 family was demonstrated by Pc 4-PDT against Jurkat cells and 5-ALA-PDT against U937 cells (human histiocytic lymphoma cells) [55,59]. Interestingly, Oka et al. identified that the accumulation of PpIX was higher in ATL leukemic cell lines compared to ATL-derived non-leukemic cell lines and immortalized normal T-cell lines by HTLV-1 infection, which was confirmed via metabolomics study, indicating progression-dependent accumulation of PpIX from the HTLV-1 carrier to ATL leukemic states [60].

### 3.3. Clinical Evidence and Clinical Trials of PDT for PCLs

PDT has exerted high selectivity in treating lymphoid malignancies in experimental studies. Sando et al. used peripheral blood mononuclear cells (PBMCs) and nucleated cells (NCCs) from 13 adult patients with lymphoid malignancies and showed that 5-ALA-PDT efficiently killed tumor cells without affecting normal lymphocytes in aggressive adult T-cell leukemia/lymphoma (ATL), while the responses of PDT on indolent tumor cells were variable [61]. Skin biopsy specimens from patients with MF also showed photodamage of Bcl-2 protein after treated with Pc 4-PDT [62].

The characteristics of photosensitizers (PSs) being used for treating PCLs are summarized in Table 2. Most PSs from in vivo case reports utilize precursors of protoporphyrin IX (PpIX) including methyl aminolevulinate (MAL) and 5-ALA, for which the two are already widely used as PSs for nonmelanoma skin cancer [10,63]. The registered clinical trials of PDT for PCLs on CinicalTrials.gov are summarized in Table 3 (accessed on 10 January 2023). The keyword terms: status: all studies; condition or disease: lymphoma; and other terms: photodynamic, were applied for first-tier filtering, and irrelevant studies about non-cutaneous lymphoma or non-PDT trials were removed. A total of six studies were found, with one study terminated.

Brumfiel et al. investigated the utility of PDT in refractory mycosis fungoides, including the refractory plaque stage and the presence of the tumor stage (NCT03281811) using a well-studied photosensitizer, ALA [64]. In this open-label study, ALA-PDT with blue light irradiation was moderately effective and well tolerated in refractory-plaque-stage MF, with response rates of 36.4% by Physician Global Assessment (PGA).

Three clinical trials (NCT00103246, NCT00023790, NCT01800838) focused on topical silicon phthalocyanine (Pc 4). Pc 4 is a second-generation photosensitizer with peak absorption in the far red at 675 nm. Baron et al. showed that a partial response rate of 40% was observed in stage I to II MF (95% CI: 0.26–0.56) by Pc 4-PDT [65]. The 0.1 mg/mL of Pc 4 and 100–150 J/cm^2^ exhibited the greatest response rate. The Pc 4-PDT was well tolerated with no significant local toxicity or increased photosensitivity.

In 2 of the 5 completed studies, the investigators focused on the application of a new photosensitizer, HyBryte™ (0.25% hypericin) ointment by Soligenix, Inc. Hypericin has maximal absorption in the yellow–red spectrum (range: 500–650 nm), penetrating up to 2 mm into the skin, and showed a potent uptake by malignant T-cells. Rook et al. conducted a dose escalation of hypericin, showing that 0.25% yielded the best therapeutic response [66]. The FLASH phase 3 randomized controlled trial (NCT02448381) showed significant efficacy after 6 weeks of treatment in early-stage MF/CTCL [67]. The study engaged 39 academic and community centers and included 196 patients aged 18 years and older with stage IA, IB, or IIA MF. Patients were randomized in a 2:1 ratio, where one group received hypericin and the other group received a placebo, respectively, to three index lesions twice weekly for 6 weeks during Cycle 1 and 6 weeks to index lesions during Cycle 2. During Cycle 3, index and additional lesions were treated for 6 weeks. The index lesion response rate (ILRR) was 16% in the treatment group vs. 4% in the placebo group (*p* = 0.04). There was a 40% increase in TLRR in patients who received two cycles of the treatment (*p* < 0.001). The most-common treatment-related adverse events (AEs) were mild local skin reactions, and no drug-related serious AEs were reported [67]. NCT05380635 further explored the safety of the HyBryte™ (0.25% hypericin) ointment by monitoring the blood level and changes of the electrocardiograms, for which the result was not posted on the website. At the time of writing (21 February 2023), the new drug application for synthetic hypericin or SGX301 (HyBryte) for the treatment of early-stage CTCL was refused by the USFDA (https://ir.soligenix.com/2023-02-14-Soligenix-Receives-Refusal-to-File-Letter-from-U-S-FDA-for-HyBryte-TM-New-Drug-Application-in-the-Treatment-of-Cutaneous-T-Cell-Lymphoma (accessed on 25 February 2023)).

**Table 2 pharmaceutics-15-01328-t002:** Photosensitizers being utilized in primary cutaneous lymphomas in in vitro studies or registered clinical trials.

Photosensitizer	Classification	Absorption Wavelength	ε (Molar Extinction Coefficient)	Depth of Penetration	Localization	Other FDA-Proved Indications
Silicon phthalocyanine 4 (Pc 4)	Porphyrin-based PS	670–770 nm (675 nm)	2 × 10^5^ M^−1^ cm^−1^	Can absorb more photons at greater tissue depth than PpIX (higher molar extinction coefficient) [65]	Mitochondria, Golgi apparatus, endoplasmic reticulum	NA
Methyl aminolevulinic acid (MAL)	Porphyrin-based PS	410 nm > 510 nm, 540 nm, 580 nm and 635 nm	5 × 10^3^ M^−1^ cm^−1^ (PpIX)	Metvix: 2 mm in BCC	Endoplasmic reticulum, mitochondria	Actinic keratosis (AK)
5-aminolevulinic acid (ALA)	Porphyrin-based PS	410 nm > 510 nm, 540 nm, 580 nm and 635 nm	5 × 10^3^ M^−1^ cm^−1^ (PpIX)	Levulan (20% solution): 1 mm in BCC Ameluz (10% gel): may be deeper	Endoplasmic reticulum, mitochondria	AK
Hypericin	Non-porphyrin-based PS	590 nm	~4.5 × 10^4^ M^−1^ cm^−1^	Up to 10 mm in animal model for colon carcinoma [68]	Mitochondria and lysosomes [69]	NA

**Table 3 pharmaceutics-15-01328-t003:** Registered clinical trials on photodynamic therapy against cutaneous lymphomas.

NCT Number Year Start	Title	Intervention	Phases	Status	Results
NCT00023790 2003	Photodynamic Therapy in Treating Patients with Skin Cancer or Solid Tumors Metastatic to the Skin	Silicon phthalocyanine 4 (Pc 4) over 2 h on Day 1, followed by light therapy over 30–60 min on Day 2. Treatments repeated 6 weeks for a total of 2 courses.	Phase 1	Terminated due to slow accrual	N/A
NCT00103246 8 February 2005	Photodynamic Therapy Using Silicon Phthalocyanine 4 in Treating Patients with Actinic Keratosis, Bowen’s Disease, Skin Cancer, or Stage I or Stage II Mycosis Fungoides	Participants receive topical silicon phthalocyanine 4 (Pc 4). One hour later, participants undergo photodynamic therapy. Treatment repeats weekly for up to 3 weeks (up to 3 total treatments for the same lesion OR up to 3 lesions treated if multiple lesions are present). Cohorts of 3 participants receive escalating doses of Pc 4 and visible light until the maximum tolerated dose (MTD) is determined.	Phase 1 44 cases were enrolled: MF 35 cases, actinic keratosis 4 cases, squamous cell carcinoma 2 cased, basal cell carcinoma, 2 cases	Completed August 2010	MF, 14/35 (40% with 95% CI: 0.26–0.56) responded [65]
NCT01800838 April 2013	Silicon Phthalocyanine 4 and Photodynamic Therapy in Stage IA-IIA Cutaneous T-Cell Non-Hodgkin Lymphoma	A dose-escalation study. Silicon Phthalocyanine 4 (Pc 4) PDT at 0.1–0.5 mg/mL with visible light at a wavelength of 675 nm at a fluence of 50–150 J/cm^2^.	Phase 1	Completed May 2015	All 11 patients completed the trial with no serious adverse events. The maximum tolerated dose (MTD) of PDT was 150 J/cm^2^, and the MTD of Pc 4 was 0.1 mg/mL.
NCT02448381 December 2015	FLASH [Fluorescent Light Activated Synthetic Hypericin] Clinical Study: Topical SGX301 (Synthetic Hypericin) for the Treatment of Cutaneous T-Cell Lymphoma (Mycosis Fungoides)	SGX301 (synthetic hypericin)-PDT for early-stage (IA-IIA) MF/CLCT, twice weekly for 6 weeks.	Phase 3 Randomized, double-blind, placebo-controlled study	Completed November 2020	Hypericin PDT was more effective than placebo, index lesion response rate (ILRR), defined as 50% or greater improvement, after Cycle 1 of treatment (16% vs. 4%; *p* = 0.04) [67]
NCT03281811 13 November 2017	Photodynamic Therapy in Treating Patients with Refractory Mycosis Fungoides	Patients receive aminolevulinic acid hydrochloride topically and undergo photodynamic therapy on Day 1. Treatment repeats every 4 weeks for up to 6 cycles in the absence of disease progression or unacceptable toxicity. Beginning at Week 24, patients undergo radiation therapy daily for 4 weeks.	Early phase 1 11 cases with 30 lesions, single group assignment	Completed 12 August 2020	Response rates of 36.4% by Physician Global Assessment [64]
NCT05380635 9 May 2022	Pharmacokinetic (PK) and electrocardiogram (ECG) Determinations Following 8 Weeks of HyBryte Treatment for Cutaneous T-Cell Lymphoma	HyBryte (0.25% hypericin, SGX301) ointment was applied to CTCL lesions and treated with visible light 18–24 h later starting at 5 J/cm^2^. Treatment performed twice a week for 8 weeks	Phase 2 Single group assignment	Completed 16 August 2022	N/A

## 4. Pros and Cons of PDT for Primary Cutaneous Lymphomas

PDT is a non-invasive, well-tolerated SDT with fewer side effects compared to steroid, surgery, chemotherapy, radiation, immunotherapy, and target therapy without specific contraindications and can be performed repeatedly in cases of the relapse of lymphomas. The average complete response rate is around 66% in early-stage MF [46].

As treatment resistance occurs in most cancer-directed treatments, it remains unknown whether intrinsic or acquired resistance to PDT in treating cutaneous lymphoma exists. An in vitro study showed that pre-incubated human histiocytic lymphoma U937 cells with a low fluence laser radiation with hypericin renders them insensitive to higher light doses [70]. Although stimulation of the expression of heat shock protein-70 (HSP-70) was observed, the exact mechanism remains to be elucidated.

A possible mechanism underlying apoptotic resistance of CTCL may be due to low expression of death receptors such as the Fas cell surface death receptor (FAS). Salva et al. proposed epigenetically enhanced PDT (ePDT) with methotrexate restoring the susceptibility of FAS-low CTCL to caspase-8-mediated apoptosis and increased response of CTCL to ALA-PDT [71].

The treatment efficacy of PDT against malignant tumors is hindered by the inherent aggregation-caused quenching (ACQ) effect of traditional PSs, the presence of antiapoptotic Bcl-2 in cells, and hypoxia in the tumor microenvironment [72]. Although Wen et al. reported systemic PDT in a leukemic animal model with increased survival, the relative hypoxic microenvironment of lymphoma offsets the possible expansion of vascular PDT [73,74]. Nevertheless, with the aid of nanotechnology, enhanced PDT with a newer photosensitizer using multifunctional hybrid nanospheres shows a promising effect in both in vitro and in vivo animal models [72,75].

## 5. Conclusions

Many case studies showed encouraging results of PDT for PLCs. The efficacy is more prominent in early-stage diseases. With the advancement of the new formulations to enhance better delivery of the photosensitizer, a longer absorption wavelength to penetrate deeper into the skin tissue, and the aid of nanomedicine to overcome a relatively anoxic tumor core [75], the treatment of PLCs with PDT is expected to have a bright future. Nevertheless, more randomized controlled trials are needed to confirm its effectiveness.

## Figures and Tables

**Figure 1 pharmaceutics-15-01328-f001:**
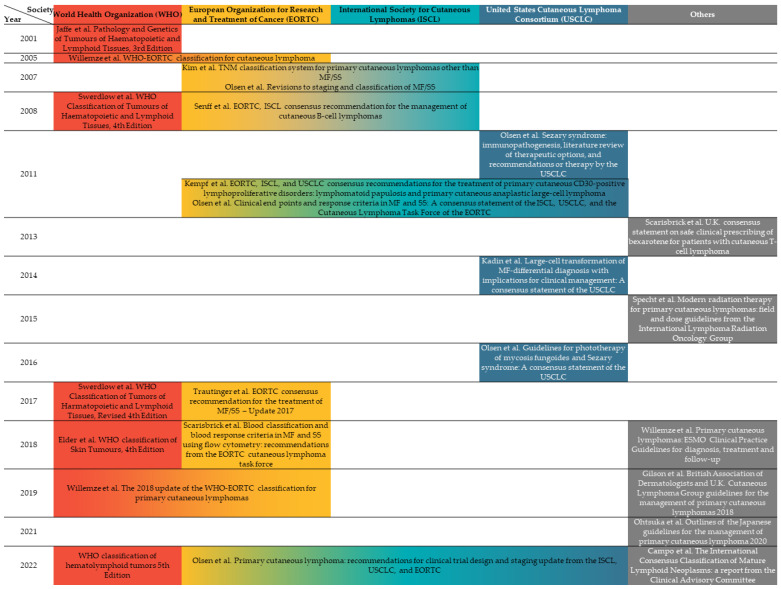
International guidelines on primary cutaneous lymphomas according to the year of publication contributed by various societies. The merged rows represent joint consensus across different societies. The color gradient shows some overlap between different guidelines.

**Figure 2 pharmaceutics-15-01328-f002:**
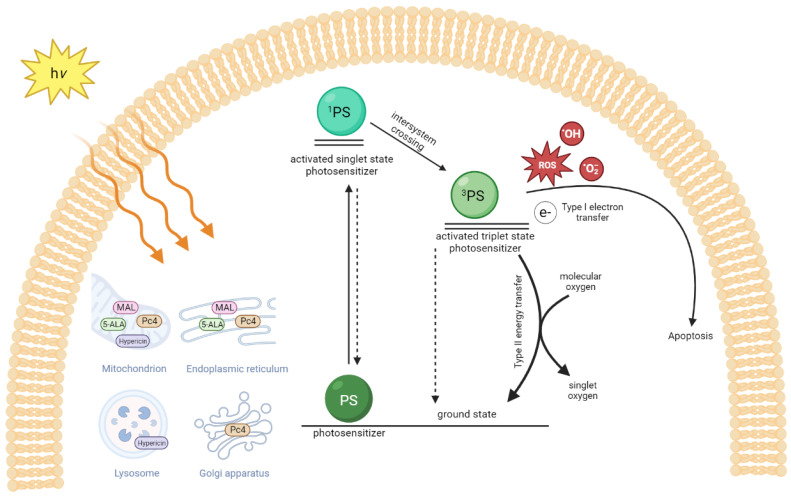
The mechanism of photodynamic therapy for primary cutaneous lymphomas. The uptake of different photosensitizers (PS) into organelles depends on the cell type and photosensitizers used. Silicon phthalocyanine (Pc 4), methyl aminolevulinic acid (MAL), 5-aminolevulinic acid (5-ALA), and hypericin are the photosensitizers used in experimental lymphoma cell studies or clinical studies. After light absorption, the photosensitizer (PS) is excited to a singlet state (^1^PS) and undergoes intersystem crossing to the excited triplet state (^3^PS). The excited triplet PS transfers its electrons to the neighbor biomolecules or oxygen to generate reactive oxygen species (ROSs), superoxide anion radical (O2•−), and hydroxyl radical (OH•) (Type I reaction) and/or by a Type II reaction through direct energy transfer to generate singlet oxygen (^1^O2) to return to the ground state PS. Figure created with BioRender.com.

**Figure 3 pharmaceutics-15-01328-f003:**
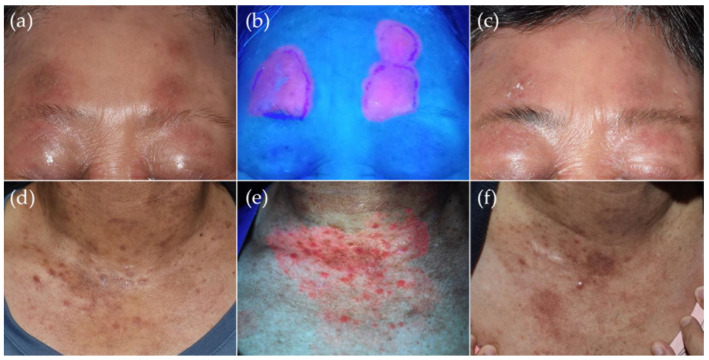
Photodynamic therapy as an adjunctive treatment in primary cutaneous lymphomas in 2 patients. A 63-year-old woman with stage IIA mycosis fungoides who received oral retinoid acid and UVB phototherapy except for the face due to the fear of skin darkening. (**a**) There are a few well-demarcated scaly erythematous patches on the forehead that responded poorly to topical treatments. (**b**) 5-aminolevulinic acid (ALA)-PDT consisted of 50 J/cm^2^ visible light (MediLED PDT, Mediland, Taoyuan City, Taiwan) irradiance after occlusion of 5-aminolevulinic acid (Ameluz, Biofrontera AG, Leverkusen, Germany) for 3 h given every other week for 3 sessions, and strong fluorescence was shown under Wood’s light examination. (**c**) After 3 treatment sessions, the erythematous scaly patches over the forehead improved greatly, although a histological exam over the right forehead lesion revealed residual atypical lymphocytes. (**d**) A 56-year-old woman with mycosis fungoides and lymphomatoid papulosis developed new crops of papules over the V-chest despite being under treatments with twice-weekly PUVA phototherapy and weekly 10 mg oral methotrexate. (**e**) Vivid coral-red fluorescence was detected under Wood’s lamp illumination after occlusion with ALA for 3 h. (**f**) Significant improvement was achieved after a single treatment with 100 J/cm^2^ visible light irradiation at 2-week follow-up.

**Figure 4 pharmaceutics-15-01328-f004:**
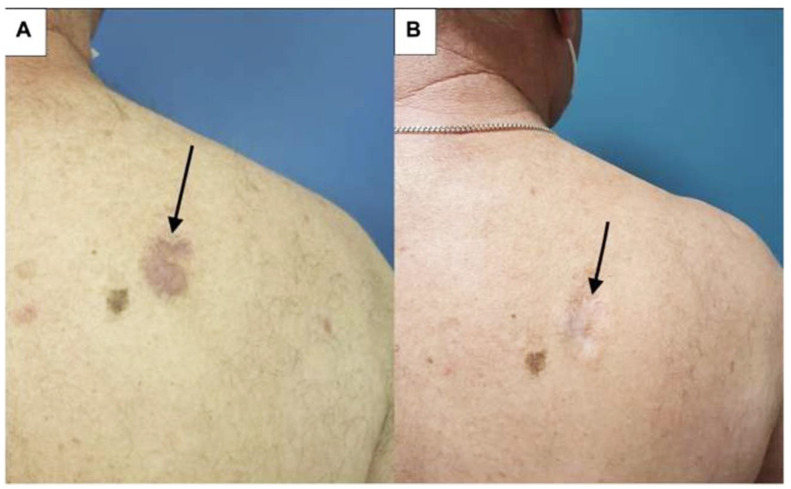
Cutaneous B-cell lymphoma of a patient (arrows pointing to the skin lesion) who received high-potency corticosteroids and 8 infusions of rituximab before PDT: (**A**) before second illumination with 37 J/cm^2^ red light after methyl-aminolevulinic acid 2.5 h occlusion and using a dermaroller to enhance drug penetration; (**B**) during follow-up showing clinical response [50].

**Figure 5 pharmaceutics-15-01328-f005:**
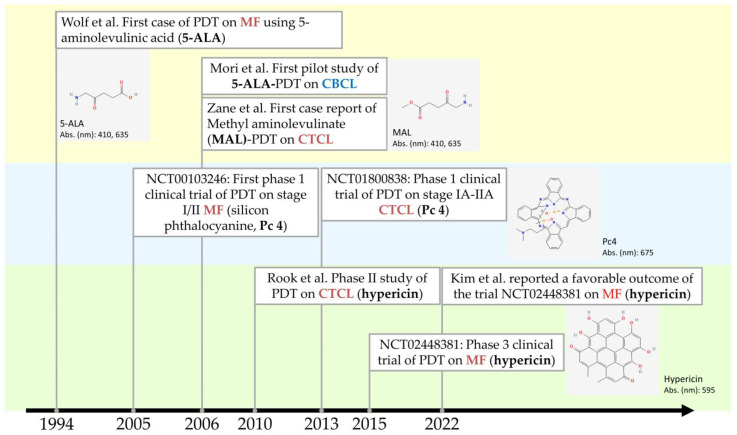
Recent advances of PDT in the treatment of primary cutaneous lymphomas. Chemical structures of photosensitizers were retrieved from the National Center for Biotechnology Information. PubChem Compound Summary for CID 3663, Hypericin; CID 119185, Silicon phthalocyanine; CID 157922, Methyl aminolevulinate; CID 137, Aminolevulinic acid. Retrieved 13 April 2023. Abs. (nm): absorption peak wavelength(s) of light of the photosensitizer.

**Table 1 pharmaceutics-15-01328-t001:** Subtypes of PCLs according to WHO-EORTC classification 2018 and changes according to International Consensus Classification (ICC) of Mature Lymphoid Neoplasms 2021 [4,19].

Entity	Disease	Changes in ICC, 2021
CTCL	MF MF variants Folliculotropic MF Pagetoid reticulosis Granulomatous slack skin SS Adult T-cell leukemia/lymphoma Primary cutaneous CD30+ LPDs C-ALCL LyP Subcutaneous panniculitis-like T-cell lymphoma Extranodal NK/T-cell lymphoma, nasal type Chronic active EBV infection Primary cutaneous peripheral T-cell lymphoma, rare subtypes Primary cutaneous γ/δ T-cell lymphoma CD8^+^ AECTCL (provisional) Primary cutaneous CD4^+^ small/medium T-cell lymphoproliferative disorder (provisional) Primary cutaneous acral CD8^+^ T-cell lymphoma (provisional) Primary cutaneous peripheral T-cell lymphoma, NOS	Primary cutaneous acral CD8^+^ T-cell LPD
CBCL	PCMZL PCFCL PCDLBLC, LT EBV^+^ mucocutaneous ulcer (provisional) Intravascular large B-cell lymphoma	Primary cutaneous marginal zone LPD

MF: mycosis fungoides; SS: Sezary syndrome; LPD: lymphoproliferative disorder; C-ALCL: cutaneous anaplastic large cell lymphoma; LyP: lymphomatoid papulosis; CD8^+^ AECTCL: primary cutaneous aggressive epidermotropic CD8^+^ cytotoxic T-cell lymphoma; PCMZL: primary cutaneous marginal zone lymphoma; PCFDL: primary cutaneous follicle center lymphoma; PCDLBCL, LT: primary cutaneous diffuse large B-cell lymphoma, leg type.

## Data Availability

No new data were created.

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
