# Peer review of "An Update on Recent Advances of Photodynamic Therapy for Primary Cutaneous Lymphomas"

_pharmaceutics, 2023, doi:10.3390/pharmaceutics15051328_

Round 1

Reviewer 1 Report

Well organized paper. It provides a good summary of our present knowledge about PDT in the treatment of primary cutaneous lymphomas and possible perspectives of this modality.

Author Response

Thank you very much for your appreciation. 

Reviewer 2 Report

In the review entitled “An update on recent advances of photodynamic therapy for primary cutaneous lymphomas” by Wong and co-workers the authors try to revise the potential of PDT to treat in primary cutaneous lymphomas.

Although the topic has interest in my opinion the review is not in conditions to be accepted. It is really hard to read and in order to improve it the authors must try to support what they say with more figures, schemes, graphics, structures of the photosensitizers in order to turn the review more appealing and didactic. They could add more figures like figure 2. The tables must be improved since they are really difficult to follow and to understand the information presented.

A review must be appealing for any reader that do not work in the field and as it is I have doubts that will have a reasonable number of citations.

Author Response

  • Although the topic has interest in my opinion the review is not in conditions to be accepted. It is really hard to read and in order to improve it the authors must try to support what they say with more figures, schemes, graphics, structures of the photosensitizers in order to turn the review more appealing and didactic. They could add more figures like figure 2.

Ans. Thank you for the important suggestions. We added more clinical figures: one case of mycosis fungoides from us who received PDT and one cutaneous B cell lymphoma case from the literature who was treated with PDT.

  • The tables must be improved since they are really difficult to follow and to understand the information presented.

Ans. Table 1 has changed into Figure 1 in which different colors were added to show guidelines from different sources for primary cutaneous lymphomas. We hope that the readers will be able to cite this manuscript not just limited to PDT treatments but also the updated classifications and guidelines in this manuscript.

Reviewer 3 Report

In this manuscript the authors made a comprehensive review on the progresses of photodynamic therapy (PDT) for primary cutaneous lymphomas. Introductions and discussions were made for different subtype of the lymphomas and their corresponding treatments. In vitro and clinical studies of PDT for primary cutaneous lymphomas were then discussed. The manuscript may be of interesting for publication by taking some suggestions into consideration.

1) As one of the most important components of PDT, it is probably better to discuss the type of lights such as wavelength, intensity and irradiation time in an individual section. Was there any requirement for lights when applying PDT for primary cutaneous lymphomas comparing with other type of skin disease? How significant lights contributed to the PDT for primary cutaneous lymphomas comparing with photosensitizers? Were there any challenges when delivering light to the treatment area (penetration depth, thermal effects, photodegradation, etc.)?

2) Comparisons should be made (or at least briefly discussed) between PDT for bacteria, fungus and cancer cells. What are the challenges and advantages of PDT for primary cutaneous lymphomas comparing with the most PDT responsive diseases?

3) The delivery of the photosensitizers and light should be included in section 3.3 beside of the dosage, treatment time and efficacy.

4) In vivo studies seem to be missing between in vitro and clinical studies. It is better to include in vivo studies before jumping into the clinical studies.

Author Response

  • In this manuscript the authors made a comprehensive review on the progresses of photodynamic therapy (PDT) for primary cutaneous lymphomas. Introductions and discussions were made for different subtype of the lymphomas and their corresponding treatments. In vitro and clinical studies of PDT for primary cutaneous lymphomas were then discussed. The manuscript may be of interesting for publication by taking some suggestions into consideration.

Ans. We sincerely thank you for your appreciation.

  • As one of the most important components of PDT, it is probably better to discuss the type of lights such as wavelength, intensity and irradiation time in an individual section. Was there any requirement for lights when applying PDT for primary cutaneous lymphomas comparing with other type of skin disease? How significant lights contributed to the PDT for primary cutaneous lymphomas comparing with photosensitizers? Were there any challenges when delivering light to the treatment area (penetration depth, thermal effects, photodegradation, etc.)?

Ans. Thank you for these critical and important questions. PDT consists of three major components: a photosensitizer, a specific wavelength of light that corresponds to the absorption peak of that photosensitizer, and oxygen. The light doses required for effective PDT on cancer are usually above 100 J/cm2 with red light while in treating benign lesions like acne or to improve wound healing are usually around 10-50 J/cm2. The following formula describes how light dose is calculated: light dose (J/cm2) = light intensity (mW/cm2) x irradiation time (seconds). A more detailed explanation of light dose, intensity, and irradiation time has been added in the Introduction.

               The limitation of light penetration, the thermal effects, and photodegradation are also added in the same paragraph.

  • Comparisons should be made (or at least briefly discussed) between PDT for bacteria, fungus and cancer cells. What are the challenges and advantages of PDT for primary cutaneous lymphomas comparing with the most PDT responsive diseases?

Ans. Thank you for these excellent suggestions. We have added the above questions, answers, and discussions in the Introduction part.

  • The delivery of the photosensitizers and light should be included in section 3.3 beside of the dosage, treatment time and efficacy.

Ans. The characteristics of photosensitizers and light are summarized in Table 3, now Table 2.

  • In vivo studies seem to be missing between in vitro and clinical studies. It is better to include in vivo studies before jumping into the clinical studies.

Ans. Thank you for your suggestion. In this review, we focused on clinical studies in humans and mechanistic studies in cells. In vivo studies were not included to avoid redundancy in the manuscript.

Round 2

Reviewer 2 Report

In my opinion the authors  did not make yet the adequate investment according with  my suggestions.  As I refer in my first report they must support their comments with  more figures, schemes, graphics, structures of the photosensitizers in order to turn the review more appealing and didactic.  Of course the additions of fig. etc from other manuscripts must have the permission of the journal (usually just in seconds they can have that permission). So considering this, I maintain my opinion that the article is not yet in conditions to atract a high number of   readers. The authors mention a lot of reviews that are presented as it is supposed to be.

About the tables  I agree that they are now much better. 

Author Response

Reviewer 2

  • As I refer in my first report they must support their comments with more figures, schemes, graphics, structures of the photosensitizers in order to turn the review more appealing and didactic.

Ans. Thank you for your suggestions and comments. We have added more figures in the revised manuscript (from 2 figures at the first submission to 5 figures now). In Figure 5, the structures of the photosensitizers were included together with the absorption peak wavelength(s) of each photosensitizer. Moreover, the uptake of different photosensitizers (PS) into organelles in a cell was added to Figure 2. We hope the revision is more readable and appealing to the readers.

  • About the tables. I agree that they are now much better.

Ans. Thank you for your appreciation.

Reviewer 3 Report

The revised manuscript has resolved the issues in original manuscript.

Author Response

  • The revised manuscript has resolved the issues in original manuscript.

Ans. Thank you for your agreement.

Round 3

Reviewer 2 Report

The authors improved slightly the manuscript and is acceptable. However   the  permission for the new figure is missing.